# The Effect of Outdoor Environmental Exposure on Readmission Rates for Children and Adolescents with Asthma—A Systematic Review

**DOI:** 10.3390/ijerph19127457

**Published:** 2022-06-17

**Authors:** Lisa Smaller, Mehak Batra, Bircan Erbas

**Affiliations:** 1Department of Public Health, School of Psychology and Public Health, La Trobe University, Melbourne, VIC 3086, Australia; lisanarelle@hotmail.com (L.S.); mehakcd@gmail.com (M.B.); 2Faculty of Public Health, Universitas AirLangga, Surabaya 60286, Indonesia

**Keywords:** asthma, child, adolescent, readmission, outdoor environmental exposure, traffic related air pollution, pollen, seasonal effects

## Abstract

The burden of asthma readmission for children and adolescents is approximately 10% worldwide. Research has been synthesised for behavioural and indoor impacts; however, no such synthesis has been conducted for outdoor environmental exposures. This systematic review aims to evaluate and synthesise the impact the outdoor environment has on readmission rates for children or adolescents with asthma. We conducted a systematic search of seven databases and hand searched reference lists of articles published up until 18 January 2021. There were 12 out of 392 studies eligible for inclusion. Overall, most studies showed that outdoor environments impact on readmission; however, the strength of association is seen to be stronger in a particular subpopulation of each study depending on the exposure investigated. The evidence for the association between outdoor environmental exposure and readmission rates for children or adolescents with asthma is increasing; however, it is complicated by potential confounders such as socioeconomic factors, ethnicity, indoor air pollutants, and other behavioural factors. Further research is required to differentiate between them. Additionally, further studies need to be undertaken in further countries other than the United States of America to understand the full relationship.

## 1. Introduction

Asthma has been described as one of the most common noncommunicable diseases in children [1,2]. Asthma has been a growing problem within the childhood population with strong effects into adulthood [1]. It is an illness characterised by inflammation leading to narrowing in the small airways of the lungs and symptoms include coughing, wheezing, chest tightness, and difficulties breathing [3]. Due to symptoms associated with asthma, it has flow on effects impacting sleep, school performance, mood, and work productivity of carers. Depending on the severity of an occurrence of asthma exacerbation, some of which may lead to a doctor’s visit or in severe cases a visit to the emergency department (ED), subsequent hospitalisation or readmission creates a public health burden. Exposure to various environmental factors can trigger asthma exacerbations. These include seasonal effects, viruses, pollen counts, and air pollution [4]. Viruses are known to cause wheezing in the paediatric population, and when combined with high levels of allergens there is an increased risk of hospital admission [5,6,7]. The International study of asthma and allergies in childhood (ISAAC) found there was a large difference in prevalence of asthma symptoms across different centres, although populations had similar genetic or ethnic backgrounds, thus proposing that environmental risk factors are the biggest determinant of prevalence within a community [8]. Many studies have been undertaken to understand factors that trigger asthma exacerbations requiring admissions; however, to a lesser extent only a handful of studies have explored risk factors of asthma readmission rates. Furthermore, very few studies [9,10,11,12,13] have investigated the impact of outdoor environmental risk factors associated with asthma readmission rates. No study has been undertaken to synthesise the outdoor environmental risk factors on readmission rates. This review synthesises the currently available literature which can be used to guide additional research, and prompts others to focus on targeted interventions or early interventions that may be undertaken to decrease the risk of readmissions due to outdoor environmental risks for children and adolescents with asthma.

## 2. Materials and Methods

The population included patients with asthma, defined as a diagnosis of asthma at readmission to hospital by using the International Disease Classification (ICD) codes, ICD-9 (493) and ICD 10 (J45 or J46). Only readmission episodes, that is where an individual has been discharged from hospital after an index admission and is readmitted to hospital within a relevant timeframe for the same diagnosis, were included within the review. There was no limit on the readmission timeframe; however. for this review the readmission timeframes have been categorised into 3 groups: within 28 days, within 6 months, and greater than 6 months. The readmission timeframes could be an indication for asthma severity or show the effect of seasonal exposures [14,15]. All articles with age groups under 18 years old were included to ensure that no articles on children/adolescents were missed. The inclusion criteria were outdoor environments. This was kept very broad; however, the focus was on outdoor air pollution. Articles for inclusion had to be published in the English language before 18 January 2021, with a study population age less than 18 years old. There were no restrictions on study design included. We conducted an updated literature search in July 2021 using the same databases and search terms. The first phase of our search strategy involved searching the peer-review literature. The applicable literature was searched in the following databases: CINAHL, Embase, Medline (Ovid), Proquest, Scopus, Web of Science, and the first 10 pages of Google Scholar. Only the first 10 pages were searched due to the search results becoming less relevant beyond that point. The search terms for each database were devised using a combination of search strings including keywords, MeSH terms, etc., and are as listed in Table 1.

The search terms were developed with the assistance of La Trobe University librarians.

All relevant articles were identified, and duplicates were removed. After this process, abstracts and titles were screened and all articles for possible inclusion were further assessed via the full text for eligibility. Furthermore, the reference lists of included articles or any review found were hand searched. The search results were managed by referencing software Endnote X7 [16].

### 2.1. Data Extraction

The data extracted from each article included first author, publication year, study population, setting, duration of study, age range, sample size, study design, readmission timeframe, exposure variable, exposure variable measurement, outcome definition according to International Classification of Disease (ICD) codes, statistical analysis, results which included parameters and a 95% confidence interval, confounders and adjustments, interactions/stratification analysis, and reporting quality. This information is in Table 2 and Table 3.

### 2.2. Assessment of Quality and Risk of Bias

A quality assessment was undertaken of included articles using the validated Zaza et al. (2000) framework [23]. This framework has been validated in the quality assessment of all types of studies and has been utilised in other reviews [24]. This framework is utilised as most study designs are not case/control or cohort, and this tool derived from Zaza et al. (2000) has been shown to perform well with ecological studies [23].

The framework is scored out of 17 by assessing a number of criteria, such as the population and eligibility criteria which were described well and were all included, whether the exposure and outcome measures are valid and reliable, whether the authors reported validated and appropriate statistical testing and if it accounted for different levels of exposure, and if the article reported limitations and appropriate controlling for confounding and other potential biases.

### 2.3. Assessment for Meta-Analysis

The exposure definitions for outdoor environments (air pollutants, pollen, seasonal trends) and the effect estimates (odds ratios, relative risks, hazard ratio, and 95% confidence intervals) were extracted from all the study findings of each article included in the review. However, due to heterogeneity of the studies (population, outcome definition (readmission timeframe), exposure definition and measurement, and measurement of effect estimates) there were no two studies alike; therefore, the results were unable to be pooled for a meta-analysis.

## 3. Results

The flow diagram (Figure 1) shows the process of review and the eligible included articles of the review. By searching electronic databases, reviewing reference lists, and omitting any duplicate articles, 211 were eligible articles for screening; however, only 129 were eligible for full-text review, and of these, 117 were excluded due to one of these listed reasons: (a) not the desired study population, (b) not related to outdoor environmental exposure, (c) did not include the outcome of readmission, (d) not related to asthma, (e) not in English, or (f) was a review. Although the reviews were excluded, the full text was read and the reference list scanned to ensure no articles were missed. After full-text review, 12 articles were included for the qualitative synthesis.

The article results have been divided into different analysis types for easier comparison within the results section.

### 3.1. Odds Ratio

There were six studies [9,10,11,12,13,22] using the odds ratio (OR) to assess the likelihood of readmission rate due to outdoor exposure. Findings for these can be found in Table 2.

All six studies considered different environmental exposure variables. Rushworth and Rob (1995) [9] focused on seasonal trends with the month of readmission used as a marker of environmental exposure to aeroallergens with the winter month used as a reference point for the analysis. Brittan et al. (2017) [10] also considered a seasonal trend of time as the exposure variable but they used pre-defined dates for each month to define the season, i.e., summer (21 June to 20 September), autumn (fall) (21 September to 20 December), winter (21 December to 20 March) and spring (21 March to 20 June). Newman et al. (2014) [11] examined traffic related air pollution (TRAP) with a focus on elemental carbon attributed to traffic (ECAT) and used as a substitute for TRAP.

Baek et al. (2020) [12] used seasonal trends; however, they also looked at the effects of air pollutants, particulate matter 2.5 (PM_2.5_), and ozone (O_3_). The seasons were divided into a warm season, defined as the months between May and October, or cold season, defined as the months between November and April. The air pollutant models were also adjusted for temperature. As Baek et al. (2020) [13] was a substudy of Baek et al. (2020) [12], the exposure is essentially the same; however, the air pollutants were both measured using the daily average prediction from the reliable United States of America environmental protection agency (U.S. EPA) Downscaler model [25], which were then divided into four categories (quartile 1 (the lowest) to quartile 4 (the highest)). In contrast, Beck et al. (2017) [17] utilised the Health and Environmental Opportunity Index (HEOI) [26] as the exposure variable. The sub-category variables for HEOI used as exposure variables are proximity from their home to parks and green space, volume of nearby toxic release, and proximity to toxic waste release sites. Each category is defined as either very low–low, medium, or high–very high.

All the studies used logistic regression analysis to assess the association between environmental exposures and readmission for asthma. In Rushworth and Rob (1995) [9], regression analysis of the month of readmission (as a marker for seasonal trends) showed 70% increased likelihood of readmission during October OR 1.70 (95% CI 1.10–2.64) and 78% increased likelihood of readmission in November OR 1.78 (95% CI 1.21–2.62). There was a 53% increased likelihood for a patient to be readmitted in spring–summer (September–December) OR 1.53 (95% CI 1.30–1.80) when compared to winter [9].

The results for the study by Beck et al. (2016) [22] found at a patient level analysis compared to population level analysis; there was a 16% increased likelihood of readmission within 12 months for the very low to low subcategory for health and environment opportunity index (HEOI) OR 1.16 95% CI 0.90–1.5, and a 3% increased likelihood of readmission within 12 months for the medium subcategory for HEOI OR 1.03 95% CI 0.72–1.46; however, once adjusted for age, race/ethnicity, and insurance, the effects became non-significant.

The analysis by Newman et al. (2014) [11] found that patients with high TRAP exposure were readmitted within 12 months at a 50% higher rate when looking at the unadjusted model *p* = 0.05 OR 1.5 95% CI 1.0–2.1; however, when the model was adjusted the results were borderline, aOR 1.4 95% CI 0.9–2.2 *p* = 0.08 [11], although overall effects of high TRAP exposure and readmission were not statistically significant [11].

Brittan et al. (2017) [10] found that a greater percentage of children with index admissions (initial admission to hospital) in summer were readmitted (39.8%) within 15–90 days post discharge, compared to no readmission (27.6%), with a significance level of *p* ≤ 0.001. Additionally, a summer discharge compared with a fall discharge had a 50% greater likelihood of readmission within 15–90 days post discharge OR 1.5 (95% CI 1.1–2.0).

The results from Baek et al. (2020) [13] indicated that most of the readmissions occurred after 90 days. In the multivariate analysis, when only individual characteristics were considered, the warm season (OR 1.51 (95% CI 1.03–2.20) *p* = 0.034) was associated with a higher number of readmissions. Additionally, children living in areas with an ozone level in quartile 2 are more likely to be readmitted within the study period than those in the lowest ozone level (quartile 1) (OR 1.78 (95% CI 1.01–3.14) *p* = 0.045 [13].

Finally, Baek et al. (2020) [12] controlled for potential confounders in their analysis through use of a time-stratified case-crossover study design. Only 8.1% of patients were readmitted within 30 days and almost 37% of all readmissions were over a year (366 days). There was a greater percentage of readmissions in the cold season (52.3%) compared to the warm season (47.7%). The analysis was conducted with single-pollutant and two-pollutant models comparing their respective effects on readmission. The two-pollutant models controlled for the other pollutant within the model [12]. The logistic regression models showed an 8.2% increased risk of readmission within the study period of 7 years with elevated PM_2.5_ concentration on Lag1 in the single-pollutant model (OR = 1.082, 95%CI 1.008–1.162, *p* = 0.030), and an 8% increased risk of readmission for the two-pollutant model (OR = 1.080, 95% CI =1.005–1.161, *p* = 0.036). Additionally, on the day of readmission (Lag0) there was a 2.3% increased likelihood of readmission due to ozone concentration levels when adjusted for PM_2.5_ (OR 1.023, 95% CI 1.001–1.045, *p* = 0.042); however, only a near significance was seen between readmission and ozone concentration Lag0 when not adjusted for PM_2.5_ (single-pollutant model) (OR 1.020, (95%CI 0.999–1.041) *p* = 0.064 [12].

In the regression models, both PM_2.5_ and ozone had a significant association with readmission only in the warm season [12]. When stratified for the warm season, the study showed elevated PM_2.5_ concentrations to have a 13.4% increased likelihood of readmission on Lag1 (OR 1.134 95% CI (1.019–1.262), a 14.6% increased likelihood of readmission on cumulative Lag0–1 (OR 1.146 95% CI (1.003–1.309) for the single-pollutant model, and a 12.5% increased likelihood on Lag1 (OR 1.125 95% CI 1.009–1.254) for the two-pollutant model [12]. This is similar when ozone concentration levels are elevated, posing a 4.3% increased likelihood of readmission on Lag0 (OR 1.043 95% CI (1.012–1.075)), 3.3% increased likelihood of readmission on cumulative Lag0–1 (OR 1.033 95% CI (1.002–1.065)), and in the single-pollutant model and the two-pollutant model elevated ozone levels increased the likelihood of readmission by 4.3% for Lag0 (OR 1.043 95% CI (1.010–1.078)) [12]. However, no significant associations were found in the cold season [12].

When stratified for age, the two-pollutant model for ozone on Lag0 showed a 2.9% increased likelihood of readmission with children aged 5–11 years old (OR 1.029, 95% CI = 1.004–1.055, *p* = 0.022) [12]. No other associations were observed between children aged 5–11 and PM_2.5_ elevated concentration levels, and no significance was found within the 12–18 years old stratification and the effects of single-pollutant or two-pollutant models on readmission rates [12].

### 3.2. General Additive Models (GAMs)

There were two studies [18,19] that used non-linear semi-parametric modelling, namely generalised additive models, to assess the functional form between exposure and readmission. Table 3 describes the characteristics and findings from these two studies.

The exposure variable considered for both studies was seasonal trends as a marker of outside environmental exposure; however, it was measured slightly differently. The Australian study [18] divided the season from hospital records into the day of the week and day of the year, whereas the Hong Kong study [19] split season into a hot season (May to October) and cold season (November to April) [18,19].

The study by Lam et al. (2019) [19] adjusted for air pollutants and other meteorological factors. Neither study tested for any interactions; however, Vicendese et al. (2013) [18] conducted data stratification for sex, and Lam et al. (2019) [19] stratified for seasonal effects.

In the hot season analysis, of the study by Lam et al. (2019) [19], there were significant non-liner effects observed for temperature and readmission with cumulative relative risk (RR 3.4 95%; CI (1.26–9.18)) and relative risk ratio (RRR of 4.59; 95% CI (1.23–17.21)) which compares the risk of readmission with first admission for asthma. Additionally, a low temperature of 10 °C was associated with readmission for asthma within the first 5 days only (RR 1.43 95% CI (1.00–2.04)) and the RRR compares the readmission to first admission 1.15 95% CI (0.74–1.81). No association was found between warm temperature and readmission rates (RR 0.88, 95% CI (0.61–1.26); and RRR 0.69, 95% CI (0.46–1.04)).

Vicendese et al. (2013) [18] conducted a chi square test of readmission rate and season which showed a very strong association *p* < 0.0005. Readmissions peaked in winter (30.5%), then autumn (28.6%), spring (24.6%), and lastly summer (16.2%). In months, the highest readmission rates were seen in June (11.75%), August (10.41%), May (10.16%), and March (10.12%). The GAM effect for modelling predictions mostly follows the actual data seasonal trends; however, the largest effects were seen in the June fortnights (25 and 26 for the year predicted) at 13.7 (95% Cl 9.4, 20.5) and 13.9 (95% CI 9.4, 20.7). Importantly, the author states they have concerns regarding reliability as the predictive lengthens, and this is seen in the wider confidence intervals.

### 3.3. Hazards Ratio

Three studies reported results from a model as a hazard ratio [20,21,22]. The characteristics and findings from these studies are described in Table 3.

The study by Beck et al. (2016) [22] examined general traffic related air pollutants (TRAP) for exposure variables. Similarly, Chang et al. (2009) [20] studied TRAP through using the ARCview geographical information system (GIS) which is an online software [27] that links addresses to three traffic proxies to account for local TRAP exposure. Although the study populations were almost the same due to Delfino et al. (2009) [21] being a sub-study for Chang et al. (2009) [20], the TRAP exposure variable studied for Chang et al. (2009) [20] was general compared to Delfino et al. (2009) [21] focusing on specific TRAP exposures; therefore, different models were used to measure the exposures. The article by Delfino et al. (2009) [21] studied a different measure of exposure using the CALINE4 dispersion models [28] which were used to estimate proxies for nitrogen dioxide (NO_2_), NOx (nitric oxide and nitrogen dioxide), and carbon monoxide at each participant’s home. Additionally, the seasonal trends were assessed as either warm season (May to October) or cool season (November to April).

According to Chang et al. (2009) [20], the effect of the three traffic-related air pollution metrics on repeat admissions (at least 8 days after index admission within the study period of 4 years) to hospital showed statistical significance for the whole population, with traffic index “residence distance (meters) to nearest arterial road or freeway” for 150–300 m compared to greater than 300 m measured as HR 1.21 95% CI 1.00–1.45, *p* = 0.05 [20]. Borderline significance was seen with the traffic index “total arterial road and freeway length within 300 m of residence” and for the whole study population with greater than or equal to 750 m. It was associated with an 18% higher rate of repeated hospital admission than children without any major roads nearby, HR 1.18 95% CI 0.99–1.41. *p* = 0.06 [20].

The study by Chang et al. (2009) [20] tested two interactions, one between sex and TRAP with no statistical significance reached and the other between insurance status and TRAP, with the effect being strongest in children with no insurance. Insurance status was used as a substitute for socioeconomic status [20]. When stratified by insurance, within the traffic index “the total length of less than 750m of major roads’’ calculated a 43% higher rate of readmission for those with no insurance, HR 1.43 95% CI (1.11–1.84) and for “the total major road length greater than or equal to 750m” a 48% higher rate of readmission for those with no insurance, HR 1.48 95% CI (1.15–1.90) [20]. For both variables, the estimates are very similar, and the confidence intervals overlap. Both individually were statistically significant at *p* = 0.05; however, when combined, were near significant with a *p* value of 0.06 when using a reference point of no major roads near the participant’s residence. Amongst the girls’ stratification, near statistical significance was seen for some individual markers of the traffic indices; however, no significance was found overall for the three traffic indices

A likely dose–response relationship for traffic density exists for those children residing within the 5th quintile of traffic density (113 VMT/day/m^3^) as there was a 21% higher repeated hospital admission recorded, HR 1.21 95% CL 0.99–1.49. Additionally, girls had a strong relationship in the 5th quintile, HR 1.49 95% CI 1.08–2.05 *p* = 0.05 [20]. Overall, there was no statistical significance for each individual trend, total major road lengths (*p* = 0.45), distance to nearest road (*p* = 0.54), or traffic density (*p* = 0.66) [20].

The study by Delfino et al. (2009) [21] conducted a hazards ratio analysis for TRAP and repeat admission within the study period of 3 years, showing a statistical significance for NOx and CO in an adjusted model (NOx aHR 1.097, 95% CI (1.034–1.164), *p* value 0.002; CO aHR 1.073, 95% CI (1.013–1.137), *p* = 0.02) [21]. However, there was no association seen for NO_2_ aHR 1.042, 95% CI (0.987–1.101). When the models were stratified by sex and TRAP association, strong statistical significance was seen for NOx and females HR 1.136 (1.043–1.238) *p* = 0.003, and an association was seen between CO and females, HR 1.1 (1.011–1.197) *p* = 0.02, but not for males. The point estimates for CO and NOx are also stronger for infants compared to older children (grouped into 1–5 years old and 6–18 years old), NOx HR 1.197 95% CI (1.075–1.333) *p* value = 0.02, and CO HR 1.158 95% CI (1.041–1.289) *p* = 0.007 [21].

The model undertaken by Beck et al. (2016) [22] undertook a different approach by first estimating the inverse probability of treatment weighting (IPTW) then fitting hazard ratio. The IPTW between the ethnicities described as “White” and “African American” for TRAP above sample mean had a standardised mean difference before IPTW of 0.390 and after IPTW of 0.038. There is no further analysis reported for TRAP.

### 3.4. Descriptive Analysis

There was one study [15] that performed descriptive analysis to assess the likelihood of readmission rate due to outdoor exposure. The study was from Victoria, Australia [10], and the total age range studied was 2 to 18 years. (See Table 3 for additional details of the design and characteristics of the studies) [15]. The study looked at two readmission outcomes: readmission within 28 days with sample size n = 2401, and readmission within 1 year with sample size n = 10,263. Vicendese et al. (2014) [15] used the date of admission to link to season to measure exposure variables, seasonal trend, and pollen.

Season was associated with readmission within 28 days for boys in autumn and summer with respective significance values of <0.001 and 0.01, whereas winter was associated with girl readmissions *p* = 0.03. Once age stratification was applied for readmission within 28 days, differences were observed for autumn and spring in the 6–12-year-old age group between boys and girls (*p* = 0.02 and *p* = 0.01, respectively). In the oldest age group, 13–18 years old, differences were observed for boys in autumn (*p* = 0.03) and girls in summer (*p* < 0.001) [15]. They also assessed grass pollen season, and this was associated only for boys *p* = 0.01 for readmission within 28 days.

### 3.5. Quality Assessment and Risk of Bias

All twelve studies received a total quality score between the range of 13.5/17 and 15.5/17 [9,10,11,12,13,15,17,18,19,20,21,22]. The quality scores for each study were scored out of 17 and possible limitations and biases are reported in Table A1. The average quality score of the twelve studies was 12.54/17. This score indicates the overall quality of the studies is fair to good.

There were several issues we found in the quality assessment of the studies. These included misclassification bias, selection bias, measurement bias, confounding bias, reporting bias, and not being generalisable across populations. These are described in detail for each article in Table A1. Misclassification bias was present in all studies due to ICD coding being used to identify asthma without checking clinical data to ensure asthma diagnosis was appropriate [9,10,11,12,13,15,17,18,19,20,21,22]. One study used a target population age of 0–5 years [19]. Additionally, a number of other studies used the category of 2 to 5 years within their study population [9,10,11,15,17,18,20,21,22]. It is difficult for asthma to be diagnosed under 5 years of age due to a number of other conditions and viruses that cause wheezing, which is the predominant diagnosis symptom [29]. The studies by Lam et al. (2019) [19], Baek et al. (2020) [13], Baek et al. (2020) [12] and Delfino et al. (2009) [21] all have an extended timeframe for the readmission definition therefore limiting the comparison of the patients’ reasons for readmission. Four studies included children with bronchodilator responsive wheezing, as well as asthma, which means the interpretation should be made with caution as wheezing due to another condition could be misinterpreted for asthma [11,20,21,22]. Beck et al. (2017) [17], Delfino et al. (2009) [21], and Chang et al. (2009) [20] not only included hospitalisation in the study population, but additionally visits to the emergency department thus limiting the interpretation of the results for comparison with other studies. Whereas Baek et al. (2020) [13] only looked at admissions but included multiple readmissions for patients (up to four readmissions). Two studies have the possibility of reporting bias for in home caregiver reported covariates [11,22]. There is also a risk of reporting bias for covariates assessed using hospital records, such as any wheezing illness being reported as asthma [21]. Additionally, widened confidence intervals in the stratified results indicate subsample sizes of n = 310 (ethnicity minority group) and n = 261 (unknown insurance status) may be limited, reducing the ability to compare the groups [21]. The Rushworth and Rob (1995) [9] and Lam et al. (2019) [19] models did not adjust for possible important individual variables such as family history of asthma, second-hand smoke exposure, and median household income. These factors have been shown to significantly impact asthma exacerbations; therefore, without adjusting for them the results may be positively skewed [9,19]. In the study by Rushworth and Rob (1995) [9], the regression analysis was not stratified by age which limits the interpretation of results due to the sample population age range being 1–44 years old, and for the purpose of this review it makes it difficult to understand the results for children.

The exposure measure used by Baek et al. (2020) [12], Baek et al. (2020) [13], Newman et al. (2014) [11], Delfino et al. (2009) [21], and Chang et al. (2009) [20] puts the results at risk of error due to population mobility. Another limitation includes one study being limited with the number of covariates used within the study due to the information coming from hospital admission data [20]. Additionally, another study did not define the population well due to description of the location not being provided, although there was a large sample used [15]. Whereas another five studies had generalisability restricted due to only one health care system being used for data collection [11,12,13,17,22]. This will restrict the findings from being transferred to other regions or countries. Finally, Brittan et al.’s (2017) [10] results should be viewed with caution for generalisation due to the sample population being limited to those who use Medicaid only.

## 4. Discussion

This is the first systematic review to look at the impact of solely outdoor environmental exposures, including air pollution, seasonal trends, and pollen effects, on readmission rates for asthma within the child and adolescent population. As a result, these findings offer a broader understanding of the complexity of the relationship between readmission rates for asthma and the outdoor environment while focusing on the gaps in knowledge to adequately compare the studies due to the limitations listed.

### 4.1. Main Findings

Our comprehensive systematic review found 12 studies undertaken in various countries around the world that met the inclusion and exclusion criteria. To understand the importance of readmission timeframe for asthma, the studies are compared within the following time frames. Firstly within 28 days (with a separate focus on those articles that had subcategories for within 28 days), then 6 months, and lastly greater than 6 months.

#### 4.1.1. Readmission within 28 Days

Two articles focused on readmission within 28 days. The study by Vicendese et al. (2013) [18] found June to be the month with the highest number of readmissions, then followed by August, May, and March. Furthermore, the month was significantly associated with trends in readmission for asthma. The month of the year can be used as a crude marker for season, and season is used as a marker of various environmental exposures, such as virus levels, so was therefore included within our review. This study indicates that season is clearly important when relating it to readmission rates. To strengthen this conclusion, Vicendese et al. (2014) [15] found that readmissions within 28 days were more strongly associated with winter. However, when stratified for age and sex, it found that readmission and grass pollen season were only associated for boys within 2- to 5-year-old and 6- to 12-year-old age groups. The link between readmission rates, grass pollen season, and boys (aged 2- to 12-year-old) has been thought to be a link between allergic reaction to pollen within this population [15]. Whereas season was linked for readmission within 28 days for boys in summer and autumn, winter was linked for girls and readmission. As Vicendese et al. (2014) [15] is a sub-study of Vicendese et al. (2013) [18], both studies use the same population; however, the statistical analysis undertaken was different in the two studies. The initial study uses GAMs, whereas the latter uses logistic regression. The evidence from these two studies suggests that season, in particular pollen season, is a risk for readmission especially for boys 6–12 years old.

A further three articles had subcategories for readmissions within 28 days for the whole study. Brittan et al. (2017) [10] found the studied population is most likely to have asthma index admissions in summer because of autumn conditions, as a very strong association was seen between repeat admission occurring when the index admission was in summer. Additionally, the patient was at a higher risk of readmission if they had a summer discharge, or if in the preceding 6 months of index admission the patient had had an oral corticosteroid prescription filled or an emergency department visit. Agreeing with these results, the study by Baek et al. (2020) [13] had readmission rates increased for those patents with an index admission during summer or fall (autumn) compared to winter. As both these studies were conducted in the northern hemisphere with a different climate, this may account for the differences in comparison to the two Australian studies [15,18]. The differences may also be due to other environmental influences such as pollen variations within each study region, virus levels within the community during the study period, or overall asthma management for an individual.

Air pollution was also important with readmission within 28 days. Ozone concentration levels measured near the patients’ residences were associated with readmission rates. Confirming the effects of ozone, the study by Baek et al. (2020) [12] also found strong effects for ozone and PM_2.5_ even after the adjustment of confounders. Elevated PM_2.5_ was significantly associated with increased readmissions within the day of readmission and the day prior. Contrastingly, ozone concentrations were only associated with readmission within the day before readmission and the readmission day when the model was adjusted for PM_2.5_. In addition, there were no differences found when stratified by sex. The findings could potentially have had a stronger effect due to a small sample size *n* = 143; a larger public-insured portion of participants within the sample confounding the results or the way the pollutants are measured may have reduced the effects. Furthermore, when strata analysis was performed for age, there was a significant association between readmission risk and ozone concentrations among 5–11-year-old patients in the two-pollutant model. This effect may be due to other factors to do with the development of the lungs of a child within this age group. Additionally, the season stratified models showed positive effects of PM_2.5_ and ozone on readmissions in the warm season but not in the cold season. It is difficult to confirm this effect of outdoor air pollutants with these two studies. It is possible that the two studies did not have sufficient power or the differences in the way exposure was measured affected the result. Additionally, the age range is more narrow than other studies which include participants aged 2 to 5.

#### 4.1.2. Readmission within 6 Months

Season was also associated with readmission within 6 months. Rushworth and Rob (1995) [9] had a 6-month readmission time frame and discovered seasonal trends were found to be present in the study with more patients readmitted during October and November (spring) in comparison to the other months of the year. The months of October to January are described as the pollen season in Australia. Additionally, significantly fewer patients who had an index admission in September had any readmission compared to the winter month index admissions. Further, patients who had an index admission in spring or early summer were more likely to have a late readmission (greater than 14 days from index admission). These findings are opposite to what other Australian studies have found with shorter readmission timeframes. This may be due to the change in climate between 1989–1990 and the other studies conducted 10 years later. An additional impact of the results, and therefore preventing comparison to be drawn, comes from the study population not specifically stratifying the age group to only include 1- to 14-year-olds.

Brittan et al. (2017) [10] also found that readmission within 15–90 days was more likely to occur if the patient had an index admission in summer. Rushworth and Rob (1995) [9] had a larger sample size, n = 782, compared to the study by Brittan et al. (2017) [10], n = 259. However, due to the differences between the readmission outcome, confounding factors not including the age range (which is expanded in Brittan et al. (2017) [10] to include 2- to 18-year-olds), and the geographical location which could impact on the seasonal impacts, it is difficult to draw a conclusive conclusion; however, it does show a pattern that summer index admission might possibly be associated with readmission within 6 months.

No study examined the impact of air pollution on readmission within 6 months other than those previously discussed above in Section 4.1.1.

#### 4.1.3. Readmission Greater Than 6 Months

The other readmission timeframes found pollen season to be associated; however, it was not associated with readmission greater than 6 months. The studies that explored the effects impacting readmission greater than 6 months only had one study [15] specifically focusing on seasonal trends, with autumn having the most readmissions compared to the other seasons; however, Vicendese et al. (2014) [15] found there was no association between grass pollen season and readmissions within 1 year. There are no comparison studies; however, the other study that focused on weather type factors was Lam et al. (2019) [19] which showed significant nonlinear effects between high temperature compared to ambient temperature and readmission. However, no association was found between warm temperature and readmission rates [19]. The participant age range of the study by Lam et al. (2019) [19] was 0 to 5 years whilst no other study included the 0 to 1 year age range. It is not possible to compare these studies as the differences in exposure measurements, study methodology, participant age range, and sample size were significantly different.

The studies that focus on air pollutants vary greatly with methodology, result analysis, sample size, and exposure measurement except for those that are sub-studies of each other. As each study measured air pollutants so differently it is difficult to compare; however, the majority showed weak to no association with the exposure variable measured.

Newman et al. (2014) [11] found ECAT as a component of TRAP and readmission within 12 months to be weakly associated when not adjusted for covariates (age, race, sex, maternal education, allergic sensitization, household income, and controller medication). However, once adjusted no association was found. The impact of the covariates indicates the significance of environmental factors, housing risks, and neighbourhood poverty on readmission rates for asthma. This is supported by Newman et al. (2014) [11] finding an interaction between race and TRAP. White children, compared to African American children, who were exposed to high levels of ECAT (a specific component of TRAP) had 3-fold higher odds of having a readmission within 12 months. Whereas African American children exposed to high levels of TRAP were not significantly associated with readmission within 12 months of index admission. This may be due to the inequalities of socio-economic situations already dividing the two ethnicities and the interaction therefore being missed.

Interestingly, the opposite was reported in Newman et al.’s (2014) [11] sub-study by Beck et al. (2016) [26], which used the same sample as Newman et al. (2014) [11], finding African American children were more likely to be readmitted and have an outdoor allergen or TRAP exposure when compared to white children in the unbalanced model. However, after balancing for TRAP and allergens, the model was found to not be statistically significant. This may mean that ethnicity is not significant when it comes to an interaction with TRAP and readmission rates, and in fact it is the socio-economic factors that impact on readmission rates. This study’s primary focus was not on outdoor environmental influence on readmission rates, instead it focused on the relationship between ethnicity and readmission. It was included in our review due to TRAP being a part of its focus on influencing factors. The broad readmission period may have also impacted on the results as it does not allow for focus on a shorter readmission timeline.

Beck et al. (2017) [17] also assessed readmission greater than or equal to 1 year using the same hospital population as Newman et al. (2014) [16] and Beck et al. (2016) [22]; however, a different sample time frame and utilisation of the health and environmental domain within the child opportunity index for the exposure variable was not significantly associated with readmissions at the patient level. The patient level adjusts for confounding for each individual compared to the whole study population. This is difficult to compare to other studies due to the exposure measurement variable not being comparable. As the population included 1–2 year-old patients who are known to be difficult to diagnose with asthma, a sensitivity analysis was performed excluding that cohort; however, the direction and magnitude of all analyses remained the same.

Chang et al. (2009) [20] focused the readmission time for their study to within 12 months. Within this sample, the greatest association of readmission and TRAP (measured from traffic metrics as a proxy for ultrafine particles, CO, and black carbon) was between female infants compared to male infants. However, 6–18 year-old females and males also had an increased rate of readmission for those that resided within 300 m of major roads. Additionally, a strong association was evident amongst children without private insurance, and some evidence of a dose–response association was seen for traffic indexes, “residence distance to nearest arterial road or freeway”, and “total major road length within 300 metres of residence”. There was a stronger effect among females and children that had no insurance or government paid insurance [20]. This reaffirms the possibility that lower income families may live in areas that have a higher level of risk factors. The sub-study of Chang et al. (2009) [20] by Delfino et al. (2009) [21] confirmed these findings with an association between TRAP and repeat admissions observed. However, the same study population was used with the only difference being that Chang et al. (2009) [20] included all encounters to hospital (admission to hospital and presentations to the emergency department) and this made their sample size slightly larger. The study by Delfino et al. (2009) also used a more focused exposure variable to assess air pollutants’ impact on readmissions and found the biggest risk for repeat admission was related to NOx (9.7% higher risk) and CO (7.3% higher risk) exposures. When the results were stratified, the strongest association was found between girls, infants, and patients living in the upper half of the income distribution. There were no associations found between distance lived from hospital and season, which may be due to the readmission timeframe not being focused on a specific timeframe and across 4 years. Nor was there association found between TRAP and insurance status; however, coefficients were larger for patients with private insurance which contradicts the study findings of Chang et al. (2009) [20], which the author investigated further and explained that those who are more likely to not have private insurance may have less stable housing situations and visited other hospitals that were not included within the study, therefore skewing the results to larger coefficients for those with private insurance [21]. Delfino et al. (2009) [21] provided a deeper assessment of the specific exposure variables and only included readmission to hospital compared to the initial study by Chang et al. (2009) [20] that included ED revisits as well as readmission to hospital.

It was also difficult to compare these studies with the Texan studies as the exposure variables, although they are also air pollutants, are different types. Baek et al. (2020) [13] and Baek et al. (2020) [12] assessed the impact of air pollutants PM_2.5_ and O_3_ on readmissions within the study period.

In addition to the different air pollutants studied, the populations across all the studies varied greatly. The Texan studies [12,13] have a 75% Latino population compared to the African American and Caucasian populations in the two Ohio studies [11,26] which excluded Latino population from their sample.

The results concluded that outdoor environmental exposures of TRAP, ECAT, Ozone, PM_2.5_, grass pollen, summer discharge, and index admissions in summer or fall showed varying levels of association with readmission rates and children with asthma. With the strongest associations seen in the 2- to 11-year-old age group, males, and Caucasian ethnicity. Nevertheless, due to the inability to appropriately compare articles due to the sample sizes, methodology differences, exposure variables examined, and outcome timeframes, it was not possible to conduct a meta-analysis of the included articles for all outdoor environmental exposures.

### 4.2. Strengths

The major strengths of our review include the incorporation of comprehensive databases for the literature search, the broad criteria for included articles, and search terms that were developed with the help of LTU librarians. The quality and assessment tool that was used to undertake quality assessment for each study is a validated and widely used tool. Furthermore, by focusing on a paediatric population for the review, the conclusions found will provide more accurate information compared to an adult population who have more comorbidities and poorer disease management.

The sample size utilised for the included studies is one of the major strengths of them. Most of the articles used large patient databases from hospitals or regional health authority record agencies. Further, most of the articles use government regulated exposure variable records, thus providing strong data for air pollutants and weather variables. One of the studies also used a time-stratified case-crossover study design which limits individual confounding and bias.

### 4.3. Limitations

Our systematic review was limited to articles in the English language which provides a possibility for some eligible studies to be missed due to language barriers. One article in the search was unable to be assessed if eligible to be included due to the language barrier [30]. Our review also includes an article that has an adult subpopulation included; however, a separate data analysis was undertaken for the child subpopulation allowing the article to be included though the findings should be used with caution in the overall interpretation of the systematic review as the data only showed rural and urban effects [9].

For the studies examining air pollutants as the outdoor environmental exposure, many of the studies used models to approximate the exposure. The models have been validated to represent the amount of exposure; however, they do not take into effect the accurate level of individual exposure for participants being studied though most studies do not use individual exposure. The models are used as an approximation of the pollutants without measuring the individual pollutant levels, making it difficult to understand the impact of each individual pollutant. To counter this effect, it might entail additional monitors being placed in high-risk local government areas. The same strategies could be undertaken for pollen exposure to understand the impact on individual types of pollen.

The inconsistencies between readmission timeframes for each study is a significant limitation. It ranges from 14 days up to 7 years. The large time range makes it impossible to compare the studies using short readmission timeframes (less than 28 days) to the studies using long readmission timeframes (greater than 1 year). The readmission timeframe has been an indication of the severity of asthma in several studies [14,15]. Therefore, this review conducted a comparison of studies within 28 days, within 6 months, and those greater than 6 months. There are only 2 studies out of the 12 that specifically look at readmissions less than 28 days [15,18] that consistently showed that month of the year can be used as a crude marker for season as an environmental factor that triggers increased readmissions in winter. It is unusual that no more studies focused on readmission within 28 days as this has been a key indicator for asthma severity. An additional three studies broke the readmission timeframe into several timeframes, one of which included less than 28 days [10,12,13]; however, these three studies found consistent results related to an increased risk of readmission if the participant had a summer or fall (autumn) index admission compared to winter. Both Baek et al. (2020) [13] and Baek et al. (2020) [12] found increased concentration levels of O_3_ within the day before readmission and the day of readmission were associated with increased risk; however, they are sub-studies of each other and no other independent study has been performed to confirm the association.

Two studies looked at readmission within 6 months [9,10]. However, the results from Rushworth and Rob (1995) [9] showed different seasonal trend results which might be limited by their extended participants’ age into adulthood impacting on the results. Five articles had readmission timeframes within the study periods of 4 years [20,21], 5 years [12,19], and 7 years [13] and did not specify the exact time frame making it near impossible for comparison to other studies. This leaves two studies that assessed the readmission timeframe of greater than 6 months [11,17], and each study used different exposures rendering it impossible to compare.

The exposure measure studied also varied greatly between studies, limiting the ability for interpretation between the outcomes and complicating meta- analysis. Four studies utilised a version of a model to approximate the level of exposure to outdoor pollution. The models included LUR, CALINE4, and Arcview GIS [11,17,20,21], but all consistently showed some level of association between outdoor air pollution and readmission rates irrespective of differences in exposure measurement. For six of the studies, season was the exposure or part of the exposure examined [10,12,13,15,18,19]. Three of those studies defined “season” the same but looked at different outdoor pollution levels and consistently showed that ozone is linked to readmission odds [12,13]. One study used patient level analysis through opportunity index census information which has not been used by any other study and is therefore not generalisable or comparable [17]. Furthermore, a number of studies used a population from one general area or city, thus limiting the generalisability of the study for other cities, regions or countries. Finally, as the studies are from many different countries and therefore represent many different populations and healthcare settings, it increases the difficulty of comparison between them. One example is Newman et al. (2014) [11], Beck et al. (2016) [22], and Beck et al. (2017) [17] which have a study population that is predominately African American or Caucasian whereas Baek et al. (2020) [13] and Baek et al. (2020) [12] have large Latino study populations.

As most of the studies were undertaken in the United States of America, further studies should be undertaken in other countries to broaden the evidence of the relationship between outdoor environmental exposures and readmission rates for children and adolescents with asthma. Additionally, supplementary studies need to be conducted in relation to specific air pollutants and pollen to gain a deeper understanding of the relationship between their effects on asthma readmissions. Monetary grants from government or private organisations would assist towards achieving these objectives.

## 5. Conclusions

In summary, our review indicates an association between environmental factors and readmission asthma, but this relationship varies greatly depending on other factors such as weather and season. It is important to continue to assess TRAP exposure and readmissions as changing climatic conditions will likely have synergistic effects with TRAP, even if TRAP is below current guidelines. In addition, future studies will also need to consider the effects of pollen during periods of heavy pollution. This may be important in countries with high levels of TRAP such as China and India. There is a high readmission burden on the individual and the public health system and it is important that these environmental exposure associations are quantified.

## Figures and Tables

**Figure 1 ijerph-19-07457-f001:**
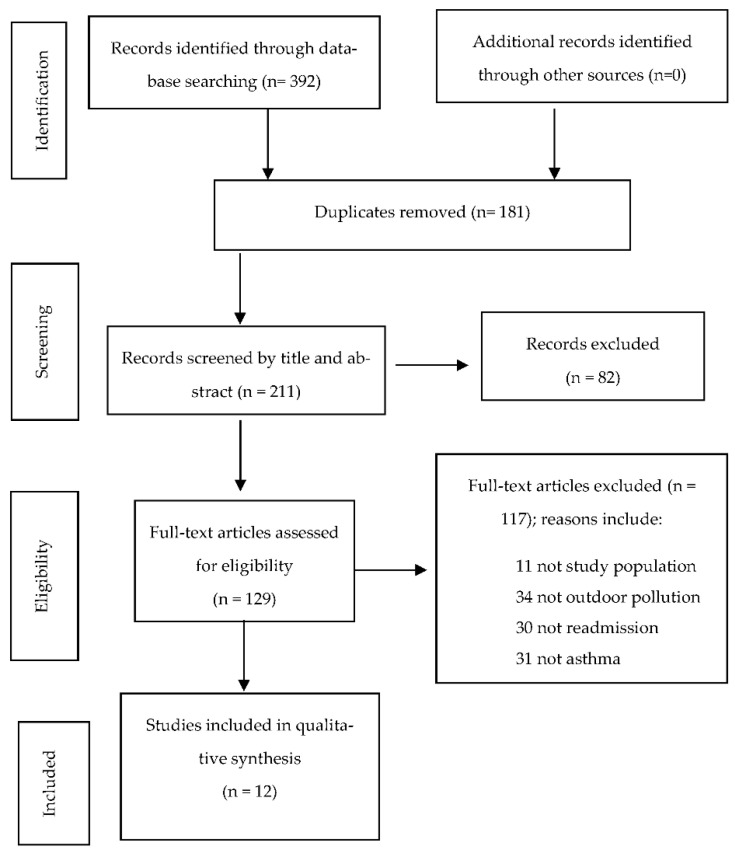
The flow diagram of the systematic review.

**Table 1 ijerph-19-07457-t001:** Search terms used for the search strategy.

Search Terms	
Environment*	1
Aeroallergen	2
“traffic related air pollution”	3
“air pollut*”	4
“Nitrous oxide”	5
“Nitrogen dioxide”	6
“Carbon Monoxide”	7
“Volatile organic compounds”	8
“Particulate Matter*”	9
Ozone	10
Pollen	11
Grass	12
Tree*	13
Weed*	14
“PM_2.5_”	15
“PM_10_” 1 OR 2 OR 3 OR 4 OR 5 OR 6 OR 7 OR 8 OR 9 OR 10 OR 11 OR 12 OR 13 OR 14 OR 15 OR 16	16 17
Child*	18
Infant	19
Youth	20
Adoles*	21
P?ediatric	22
“young people”	23
Teen*	24
18 OR 19 OR 20 OR 21 OR 22 OR 23 OR 24	25
Readmi*	26
“repeat admi*”	27
Re-hospitali?ation	28
“repeat hospital encounters”	29
26 OR 27 OR 28 OR 29	30
Asthma*	31
17 AND 25 AND 30 AND 31	32

**Table 2 ijerph-19-07457-t002:** The characteristics and findings of included Odds Ratio studies.

Author(s) (Year)	Study Population /Setting/Duration/Age of the Participants/Sample Size/Design of the Study	Exposure Variable/ Exposure Measurement	Readmission timeframe/ Outcome Definition/ Statistical Analysis	Confounders/Covariates/ Interactions	Results	Findings
Rushworth et al. (1995) [9]	Patients (1–14 years) admitted for asthma (ICD9-CM code 493) to all New South Wales (NSW) hospitals including NSW residents who attended hospital interstate for the financial year 1989–1990 and first 2 weeks for 1990/1991. Readmitted included n = 782. Descriptive study.	Months of the year were used to assess the seasonal effect. Season is a crude marker of environmental effects of pollen and viruses. Winter months (July to August are used as the reference point for the results analysis).	Readmission within 6 months of index admission. Only the first readmission from the index admission was counted. Descriptive analysis.	All regression models adjusted for sex, age, geographic location (rural/inner metropolitan/outer metropolitan), length of hospital stay in days, type of hospital (teaching/non-teaching hospital). No interactions were fitted in the regression models. Strata analysis was performed for sex, urban (inner/outer), and rural and for hospital type.	The effect of month of readmission was statistically significant during October and November: OR 1.70 (95% CI 1.10–2.64) and 1.78 (95% CI 1.21–2.62), respectively. The OR in spring–summer (September–December) was 1.53 (95% CI 1.30–1.80) when compared to winter.	Seasonal trends were found with more patients readmitted during October and November (spring). Significantly fewer patients who had an index admission in September had any readmission compared to the winter month index admissions. Patients who had an index admission in spring or early summer were more likely to have a late readmission (greater than 14 days from index admission).
Newman et al. (2014) [11]	Patients (1–16 years) admitted for asthma or bronchodilator-responsive wheezing (ICD-9 (493.XX or 786.07)) to the Cincinnati Children’s Hospital Medical Centre (CCHMC), an urban tertiary care hospital, between August 2010 and October 2011; also included nearby satellite inpatient facility from November 2010. The readmission sample size, n = 774 Descriptive study.	Traffic related air pollution (TRAP) measured as elemental carbon attributed to traffic (ECAT). They used modelling of a validated Land use regression model (LUR) (to reported residence) as a substitute to measure TRAP. ECAT was measured at the median level of 0.37 microgram/m^3^ and either classified as above or below this result.	Readmission within 12 months of index admission. Logistic regression analysis and cox proportional hazards models.	The modelling adjusted for sex, age, tobacco exposure, race (African American or white), housing risks, neighbourhood poverty, socioeconomic factors, outdoor allergen factors, indoor allergen factors, asthma controller use, and maternal education. This study assessed an interaction between race and Trap.	Patients with high TRAP exposure were readmitted at a higher rate: OR 1.5 (95%CI 1.0–2.1), *p* = 0.05 aOR 1.4 (95%CI 0.9–2.2) TRAP and race interaction: OR 1.5 (95% CI 0.9–2.5), *p* = 0.07. When broken into individual race: white-OR 3.0 (95%CI 1.1–8.1) *p* = 0.03 African American-OR 1.1 (95% CI 0.6–1.8) *p* = 0.7.	TRAP and readmission within 12 months are weakly associated when not adjusted for covariates; however, once adjusted no association was found. This study did find an interaction between race and TRAP. White children who were exposed to high levels of TRAP had a 3-fold higher likelihood of having a readmission within 12 months. Whereas African American children exposed to high levels of TRAP were not significantly associated with readmission within 12 months of index admission.
Beck et al. (2017) [17]	Patients (1–16 years) living within Hamilton County, Ohio, with a hospitalisation or visited Emergency Department for asthma or wheezing (ICD-9, (code 493.xx)) at the Cincinnati Children’s Hospital Medical Center (CCHMC) between January 2011 and December 2013. Sample size n = 1845 Descriptive study.	The subcategory variables to the health and environmental opportunity index (HEOI) used as exposure variables are proximity from their home to parks and green space, volume of nearby toxic release, and proximity to toxic waste release sites. For patient level analyses, each opportunity index was measured in categories, i.e., very low, low, medium, high, or very high, to represent quintiles of the census tract z-scores.	Hospital readmission within 12 months of the index admission for the study period and the patients’ address was geocoded and mapped to the in-county census tract.	The analysis was adjusted for age, race/ethnicity (Hispanic, white, African American, or other), and insurance (private or public). There were no interactions in this study.	At a patient level analysis for HEOI, the very low–low category: OR 1.16 95%CI (0.90–1.50), aOR 0.98 (0.75–1.29); At the moderate level: OR 1.03 (0.72–1.46) aOR 1.1 (0.71–1.44), *p* = 0.20	The environment domain was not significantly associated with the outcome at the patient level. As the population included 1–2-year-old patients who are known to be difficult to diagnose with asthma, a sensitivity analysis was performed excluding that cohort.
Brittan et al. (2017) [10]	Patients (2–18 years) hospitalised for asthma (ICD-9 (493.xx)) between 1 July 2009, and 30 June 2011, and continuously enrolled in Medicaid from 1 of 12 states for 6 months before and 3 months after the hospitalization with health claims contained in the Truven Health MarketScan Database (Ann Arbor, Michigan). The readmitted sample size was 259. Descriptive study.	The effect of seasonal trends. This study defined each season as the inclusive dates below: summer (21 June to 20 September), autumn (fall) (21 September to 20 December), winter (21 December to 20 March), and spring (21 March to 20 June).	Hospital readmission from 15 to 90 days after discharge from the index admission within the study period. For those children with greater than 1 readmission, 1 hospital admission was randomly chosen. Multivariable logistic regression models.	The modelling adjusted for sex, age, season, Medicaid, disability, and comorbid complex chronic condition. There were no interactions in this study.	Of the children who were readmitted, 35.5% were aged 2–4, 45.6% 5–11 years old, and 18.5% aged 12–18 years. The median number of days between discharge from index admission and readmission was 50 days with inter quartile range of 32–70. Summer discharge compared to fall (autumn) discharge had a greater likelihood of readmission OR 1.5 95% CI (1.1–2). Greater percentage of children with index admission in summer had readmission (*p* ≤ 0.001).	The studied population is most likely to have asthma readmissions in autumn because of summer discharge. The patient was at a higher risk for readmission if they had a summer discharge, or if in the preceding 6 months of index admission the patient had had an oral corticosteroid prescription filled or an emergency department visit.
Baek et al. (2020) [13]	Patients (5–18 years) readmitted to Driscoll Children’s Hospital, South Texas, for Asthma (ICD-9 or ICD-10) between 2010 to 2016. Total readmissions were 143. (2–3 admissions n = 121 4 admissions n = 22). Ecological time series study.	The air pollutants (PM_2.5_ and ozone) were both measured by using the daily average prediction using the Downscaler model of the U.S. EPA at census tract level, then divided into four categories as quartiles. Seasons were defined as warm, from May until October, or cold, from November until April.	Readmission timeframes were split into 1–30 days, 31–90 days, 91–365 days, and 366 days or longer. This study utilised bivariate and multivariate logistic regression (3 models created (unadjusted, adjusted model with only individual-level factors, and adjusted model with individual-level and environmental factors)).	The study analysis controlled for age (5–11 years old or 12–18 years old), length of stay (LOS), season (warm or cold), type of insurance (public or private), sex, year, ethnicity (Hispanic or non-Hispanic), and use of medication. There was no formal interaction analysis conducted.	Most of the total readmissions were after 90 or more days. None of the environmental factors showed significant differences between the different readmission groups; however, quartile 4 of the ozone group waslarger in high utilisation groups (*p* = 0.052). In the multivariate analysis, when only individual characteristic was considered, season (warm, *p* = 0.034) was associated with readmission. When environmental factors were considered, very similar associations were seen. Children living in areas with an ozone level of quartile 2 were more likely to be readmitted than those in the lowest ozone level (*p* = 0.045).	Readmission rates are increased for those patents with an index admission during summer or fall (autumn) compared to winter. Ozone concentration levels measured near the patients’ residence were associated with readmission rates. However, air pollutants and social vulnerability index were not highly correlated
Baek et al. (2020) [12]	Patients (5–18 years) readmitted to Driscoll Children’s Hospital, South Texas, for Asthma (ICD-9) between 2010 to 2014. Sample size = 111 Ecological time series study.	The daily mean concentration of PM_2.5_ was measured in micrograms per cubic metre (μg/m^3^) and ozone was measured by the mean 8-h average concentration in parts per billion (ppb). Temperature data was collected from the air monitoring stations that were closest to the patients’ home address. Seasons were defined as warm, from May until October, or cold, from November until April.	Readmission within the study period.	The analysis controlled for age (5–11 years old or 12–18 years old), length of stay (LOS), seasonal effect, type of insurance (public, private or self-paid), sex, year, and ethnicity (Hispanic or non-Hispanic) by using a case-crossover study design. However, temperature was adjusted for the analysis. There was no formal interaction analysis. Strata analysis was conducted between the conditional logistic regression analysis and age, sex, and season.	Only 8.1% of patients were readmitted within 30 days, and almost 37% of all readmissions were over a year (366 days). There were a greater percentage of readmissions in the cold season (52.3%) compared to the warm season (47.7%). Both PM_2.5_ and ozone had a significant association on readmission only in the warm season. Increased risk of readmission with elevated PM_2.5_ concentration.	Strong effect for ozone and PM_2.5_ even after adjustment of confounders. Elevated PM_2.5_ was significantly associated with increased readmissions in the short term only. Whereas Ozone was only associated in the short term when the model was adjusted for PM_2.5_. There were no differences found when stratified by sex. However, when strata analysis was performed for age, there was a significant association between readmission risk and ozone concentrations among 5–11-year-old patients in the two-pollutant model. The season stratified models showed positive effects of PM_2.5_ and ozone on readmissions in the warm season but not in the cold season.

OR = Odds Ratio, aOR = adjusted Odds Ratio.

**Table 3 ijerph-19-07457-t003:** The characteristics and findings of included GAM, Hazard Ratio, and Descriptive analysis studies.

Author(s) (Year)	Study Population /Setting/Duration/Age of the Participants/Sample Size/Design of the Study	Exposure Variable/ Exposure Measurement	Readmission Timeframe/Outcome Definition/Statistical Analysis	Confounder/Covariates/ Interactions	Results	Findings
GAM studies						
Vicendese et al. (2013) [18]	Daily childhood asthma hospital admissions (2–18 years) with a principal diagnosis of asthma, ICD-9 codes (493) up to 1998 and ICD-10 codes (J45 or J46) in Victoria, Australia, between 1997 and 2009 from the VAED Department of Human Services. n = 2401 Study length—13 years Ecological time series study.	Seasonal trends. The hospital records were divided into day of week, month of year, and day of year. The month variable was used to measure the seasonal trend.	Patients were included if their readmission was within 28 days from their index admission. Hospital readmission with a principal diagnosis of asthma, ICD-9 codes (493) up to 1998 and ICD-10 codes (J45 or J46) for remaining data to 2009 for the study period. GAM analysis.	The modelling was adjusted for the covariates including season (month), day of week, and day of year. There were no interactions measured. However, a data stratification was done for sex.	The chi square test conducted for readmission rate and season showed a very strong association with a *p* value of <0.0005. Readmissions peaked in winter (30.5%), then autumn (28.6%), spring (24.6%), and lastly summer (16.2%). Regarding months, the highest readmission rates were seen in June (11.75%), August (10.41%), May (10.16%), and March (10.12%). Both semi parametric and non-parametric models showed that both month and day of the week had significance to the overall daily readmission rates.	The study found June to be the month with the highest number of readmissions, then followed by August, May, and March. Furthermore, the day of the week and month were significantly associated with trends in readmission for asthma.
Lam et al. (2019) [19]	Patients (0–5 years) admitted for asthma (ICD-9 code 493.xx) in all public hospitals in Hong Kong between 2002 and 2011 from the Hospital Authority in Hong Kong. Ecological time series study. There were 2185 readmissions.	Seasonal trends. The seasons were defined as hot season (May to October) and cold season (November to April). The other meteorological measures included daily mean temp (°C), daily mean relative humidity (%), daily mean wind speed (km/h), daily total solar radiation (J/m^2^), and daily rainfall (mm).	Readmission within the study period. A general additive models (GAMs) analysis and Distributed Lagged Nonlinear models (DLNMs) were used for lagged effects and other nonlinear associations.	The analysis was adjusted for seasonal patterns, day of the week effects, air pollutants and other meteorological factors. There were no interactions tested Stratified for seasonal effects.	Significant GAMS were observed for temperature and readmission. The hot season analysis showed a relative risk (RR) of 3.4 (1.26–9.18) and a relative risk ratio of (RRR) 4.59 (1.23–17.21). The cold season analysis with 10 °C vs. 15 °C showed RR 1.43 (1.00–2.04) and RRR 1.15 (0.74–1.81), and with 21 vs. 15 °C, RR 0.88 (0.61–1.26) and RRR 0.69 (0.46–1.04).	High temperatures were strongly associated with asthma readmission; however, no association was found between warm temperature and readmission. Low temperatures were associated with a risk of readmission; within the first 5 days had the strongest effect.
Hazard Ratio studies						
Chang et al. (2008) [20]	Patients (0–18 years) admitted for asthma as primary or secondary diagnosis to the Children’s Hospital of Orange County (CHOC) and the University Children’s Hospital of the University of California Irvine Medical Center (UCIMC), between 1 January 2000 and 31 December 2003. Repeat admissions were defined as 2 or more encounters. n = 817. Ecological time series study.	TRAP was represented by 3 traffic proxies calculated by measuring traffic density by measuring distance to major roads surrounding the participants’ house. No specific pollutant was measured.	All readmissions for asthma after 8 days from index admission during study period.	All regression models were adjusted for age, sex, race (white non-Hispanic, white Hispanic, black, Asian, other, or unknown), insurance status (private, government sponsored/self-paid, or unknown), distance of participant’s home residence from treating hospital, and their median household income. Two interactions were tested in this study. The first test was between sex and TRAP, and the second interaction was between insurance status and TRAP.	The time to first readmission included within 1–2 months (26.5%), 3–6 months (26.9%), 7–12 months (21.6%), and after 1 year (22.1%). Traffic-related air pollution metrics on repeat admissions to hospital showed, for 150–300 m compared to greater than 300 m, HR 1.21 95% CI 1.00–1.45, *p* = 0.05. Additionally, “total arterial road and freeway length within 300 m of residence” and whole study population with greater than or equal to 750 m was associated with an 18% higher rate of repeated hospital admission than children without any major roads nearby, HR 1.18 95% CI 0.99–1.41. *p* = 0.06. When stratified by insurance, within that index the total length of less than 750 m of major roads calculated a HR 1.43, 95% CI (1.11–1.84), and for the total major road length greater than or equal to 750 m HR 1.48, 95%CI (1.15–1.90). Girls who had greater than 750 m of total arterial road length within 300 m of their house had a 39% higher rate of readmission compared to none, HR 1.39, 95% CI (1.04–1.87).	Over ¾ of the sample were readmitted within 12 months, with ¼ of the sample readmitted within 2 months and ⅓ readmitted with 3–6 months. Within this sample, the greatest association of readmission and TRAP was between female infants compared to male infants. However, 6–18-year-old females and males also had an increased rate of readmission for those that resided within 300 m of major roads. Some evidence of a dose–response association was seen for traffic indexes, residence distance to nearest arterial road or freeway, and total major road length within 300 metres of residence.
Delfino et al. (2009) [21]	Patients (0–18 years) admitted for asthma (ICD-9 493) to the Children’s Hospital of Orange County (CHOC) and the University Children’s Hospital of the University of California Irvine Medical Center (UCIMC), between 1 January 2000 and 31 December 2003. The sample size was defined as 2 or more admissions, n = 697.	Nitrogen dioxide (NO2), Nox (Nitric oxide and Nitrogen dioxide), and carbon monoxide (CO) concentrations were estimated as the local traffic emissions for both trucks and vehicles within a 5 km radius of each residence. The exposure was stratified by either warm season (May–October) or cool season (November–April). TRAP was measured using CALINE4 model estimates.	Readmission within study timeframe of 4 years or by the patient’s 19th birthday. Only 10 readmissions or less per patient were included. Time to event. Regression analysis using Hazards ratio to estimate effects of environmental factors.	All regression models were adjusted for sex, age, health insurance (private, government sponsored/self-paid, or unknown), census-derived poverty, median household income, race/ethnicity (white non-Hispanic, white Hispanic, black, Asian, other, or unknown), residence distance to hospital, and season. Strata analysis was undertaken for sex and age group. There were no interactions found.	The hazards ratio analysis for TRAP and repeat admission (NOx HR 1.094 (1.035–1.156) *p* = 0.002, aHR 1.097 (1.034–1.164) *p* = 0.002, CO HR 1.072 (1.016–1.131) *p* = 0.01, aHR 1.073 (1.013–1.137) *p* = 0.02). However, there was no association seen for NO2: HR 1.044 (0.992–1.098) *p* = 0.10 and aHR 1.042 (0.987–1.101). When the models were stratified for sex and TRAP association, for NOx and females HR was 1.136 (1.043–1.238) *p* = 0.003 and an association was seen between CO and females with HR 1.1 (1.011–1.197) *p* = 0.02. The point estimates for CO and NOx are also stronger for infants compared to older children: NOx HR 1.197 95% CI (1.075–1.333) *p* value = 0.02 and CO HR 1.158 95% CI (1.041–1.289) *p* = 0.007.	This study found an association between TRAP and repeat hospital admissions. The biggest risk for repeat admission was related to NOx and CO exposures. When the results were stratified, the strongest association was found between girls, infants, and patients living in the upper half of the income distribution. There was no association found between distance lived from hospital and season.
Beck et al. (2016) [22]	Patients (1–16 years) admitted for asthma or bronchodilator-responsive wheezing ICD-9 (493.xx) at the Cincinnati Children’s Hospital Medical Center (CCHMC), an urban, academic, pediatric hospital, between August 2010 and October 2011. n = 774 total population n = 695 readmission population. Descriptive study.	Traffic related air pollutants (TRAP) (Elemental carbon attributed to traffic). TRAP was estimated using land use regression (LUR) models. TRAP was measured as the average daily elemental carbon attributed to traffic (ECAT). When the measure was developed, it was further classified as above or below the median level of 0.37 microgram/m^3^.	Readmission within the study period of 14 months. Weighted cox proportional hazard regression and Kaplan–Meier curves.	The analysis was adjusted by age, sex, ethnicity (African American or white), tobacco exposure, insurance status, vehicle ownership, primary care access, socioeconomic status, caregiver education level, allergen sensitisation, indoor allergens, and outdoor allergens. There were no interactions in this study. The allergens were stratified.	The inverse probability of treatment weighting (IPTW) between “White” and African American” for TRAP above sample mean had a standardised mean difference before IPTW was 0.390 and after IPTW was 0.038. For any outdoor allergen sensitisation, the standardised mean difference before IPTW was 0.313 and after was 0.009.	In summary, African American children were more likely to be readmitted and have an outdoor allergen or TRAP exposure (approximated from LUR) when compared to white children. After controlling for TRAP and allergens in general, the model was found to be not significant.
Descriptive analysis						
Vicendese et al. 2014 [15]	Daily childhood (2–18 years) asthma hospital admissions ICD-9 (493) up to 1998 and ICD-10 (J45 or J46) for remaining data until 2009 in Victoria, Australia, between 1997 and 2009 from the Department of Human Services. Readmissions within 28 days were 2401, and readmissions within 1 year were 10263. Ecological time series.	Seasonal trends and the effect of grass pollen season between October and January. The hospital records were divided into autumn, winter, spring, and summer based on the month of admission. The grass pollen season was defined as between October and January, as per Burkhart TRAP.	Readmission within 28 days and 1 year from index admission. Logistic regression.	The analysis was adjusted for age, sex, season, and grass pollen season. There were no interactions measured. A data stratification was done for sex and age in categories (2–5, 6–12, and 13–18 years).	For readmission within 28 days, season (*p* < 0.001) and age (*p* = 0.01) were associated for boys in the autumn and summer seasons and age were significant (*p* = 0.03), and in winter for girls. Once age stratification was applied, differences were observed for autumn and spring in the 6–12-year-old age group between boys and girls (*p* = 0.02 and *p* = 0.01, respectively). In the oldest age group, 13–18 years old, differences were observed for boys in autumn (*p* = 0.03) and girls in summer (*p* < 0.001). Grass pollen season and age group were only associated for boys (*p* = 0.01). Whereas readmission within 1 year and summer were associated with boys *p* = 0.04 and winter *p* = 0.03 and summer *p* = 0.004 for girls. However, readmission within 1 year and the grass pollen season for both girls and boys (*p* = 0.94 and *p* = 0.31, respectively) were not associated.	Readmissions within 28 days were strongly associated with winter for girls and autumn and summer for boys. When stratified by age and sex, readmissions within 28 days and grass pollen season were only associated for boys. No association was found between grass pollen season, age group, or sex for readmissions within 1 year.

OR = Odds Ratio, aOR = adjusted Odds Ratio.

## Data Availability

Not applicable.

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
