# Peer review of "The Effect of Outdoor Environmental Exposure on Readmission Rates for Children and Adolescents with Asthma—A Systematic Review"

_ijerph, 2022, doi:10.3390/ijerph19127457_

Round 1

Reviewer 1 Report

This manuscript by Smaller et. al. and a review of the literature surrounding the effects of outdoor environment exposure on readmission rates for children and adolescents with asthma indicate that there is a relationship. The relationship varies greatly depending on weather factors and the individual outdoor environmental factors. Ozone, PM2.5, NO2, and NOx all have an impact on asthma readmission rates however to a varying degree. The whole manuscript was well-organized. Hence, I recommend its publication in IJERPH after a minor revision with the following comments addressed. 

1. Pay attention to how to write NO2 rather than NO2.

2. The CONCLUSION is perhaps the only weakest part of the review - it is a little generic. It would be nice to have a few more thoughts from the authors about challenges and opportunities in this area for the future.

3. There are a few areas where the English could be improved, such as some past and present tense.

4. There are some grammatical errors in this manuscript such as continuously forgetting to add ‘a’ or ‘the’ before a specific word which limits the clarity of the author’s writing

This review addresses the strengths and limitations of the systematic review namely the comprehensive database search and the inconsistencies between readmission timeframes. It is important going forward into the future that the knowledge gaps have been identified and therefore will be able to be addressed including a definition for the focus of readmission time frame, a need for further research within Australia, and the impact of TRAP and pollen counts on readmission rates and ways to help mitigate the effects on asthma readmissions.

Reviewer 2 Report

I read the review article by Smaller, Batra and Erbas with great interest.

In general, they give a comprehensive summary of the currently available studies dealing with the the possible impact of different outdoor environmental factors on asthma readmission rates in children and adolescents. Nevertheless, I have a few minor comments and suggestions for improvement:

1 Lines 47+48: Please cite the “very few studies“ that have investigated the impact of outdoor environment risk factors.

2 Table 1: Please add one column with numbers of the search terms so that the according search terms can be found back more easily when they are mentioned as “19 OR 20 OR 21 OR 22 OR 23 OR 24 OR 25“ etc.

3 Table 1: since you mention the “La Trobe University librarian“several times in your manuscript, maybe you want to mention their name in the acknowledgement section?

4 Figure 1: Why do you mention the last square (Studies included in quantitative synthesis (meta-analysis) (n =0)) in this figure? I would leave it out and just have it mentioned in the text, since you didn’t have any suitable studies for it.

5 Table 2: Since you mention and describe all the studies in your main text, I would suggest shortening the text in the table more. If possible (maybe depends on the journal’s formatting rules) also change the formatting so that the info fits at least just on one page per study. Now, it is cumbersome to read.

6 Table 2, study of Baek et al, 2020, Study population: I don’t understand what the percentages in brackets at the admissions mean. If 2-3 admissions are n=121 (13,4%) and 4 admissions are n=22 (2,4%), what are the other 84,2%? And if there is a total of 143 readmissions, why is the n-number of 121 only 13,4%? Please explain and rephrase in the table.

7 Table 3, study of Lam et al, 2019: Findings: Please delete the crossed “However“

8 Lines 382+505: I suggest to use “our“ instead of “my“ as you are three authors.

9 Appendix A: Please add a column on the strengths you see in each study. This will further guide researchers of future studies to orient themselves on how to design and conduct their studies on this topic.
